# OpenReview forum: "Beyond Static Allocation: Dynamic Sensitivity-Aware Fine-Tuning for Vision Transformers"
_ICML.cc/2026/Conference — ICML 2026 regular_

### Official Review · Reviewer_it7i · 2026-02-14

**Soundness:** 3
**Presentation:** 3
**Significance:** 3
**Originality:** 3
**Overall Recommendation:** 4
**Confidence:** 4

**Summary:**

Most existing PEFT methods for ViT are statically configured, leading to poor optimization and redundant activation parameters. To address this, the manuscript proposes a Dynamic Adaptive Fine-tuning (DAF) framework, which dynamically adjusts and reconfigures the trainable structure through a Perceive-Decide-Execute cycle strategy. DAF achieves state-of-the-art results on multiple benchmarks and various vision tasks.

**Compliance With Llm Reviewing Policy:**

Affirmed.

**Final Justification:**

The rebuttal improves my confidence in the manuscript. I will maintain my current score.

**Key Questions For Authors:**

1, DAF treats each layer's sensitivity independently, ignoring potential cross-layer relationships. An inter-layer analysis is a necessary clarification.


2, In the related work section, it is a lack of discussion for the latest developments in PEFT, especially since some VPT methods [1-3] in 2025 have already focused on cross-layer relationships in ViT.

[1] Sharing Task-Relevant Information in Visual Prompt Tuning by Cross-Layer Dynamic Connection

[2] Visual Prompt-Agnostic Evolution

[3] Exploring Interpretability for Visual Prompt Tuning with Cross-layer Concepts


3, The claimed contributions appear overstated, DAF is specifically designed around dynamic LoRA allocation. It may not generalize to other PEFT methods, such as adapters and prompt tuning.


4, The ratios among Matrix, Vector, and Special are critical hyperparameters. It is necessary to provide ablation.


5, The main text lacks essential visual analysis to support the core claims. For instance, a visualization of how DAF evolves across reconfiguration cycles is a good choice that verify the motivation.

**Limitations:**

yes

**Strengths And Weaknesses:**

Strengths:
Different layers in ViT learn different information and patterns, the limitations of the static assumption make sense. The proposed DAF dynamically adjusts the PEFT budget during training to match the optimization objectives, which is more efficient than static allocation. And adaptively deciding which blocks and operations require activation using gradient information is more efficient than static allocation. Multiple blocks with 6 operation positions offer flexible configuration.

Weaknesses:
Although DAF is applied to the different layers, it lacks a discussion or analysis of the inter-layer relationships. At its core, the method is a dynamic LoRA strategy that periodically reallocates where LoRA modules are inserted. The budget ratios for the three activation parameters appear to be an empirical setup.

---

> ### Author Rebuttal · Authors · 2026-03-31
>
> We sincerely thank Reviewer it7i for the constructive comments and helpful suggestions. We address each point below.
>
> >  **Response to W1 & Q1 (Inter-Layer Relationships)**
>
> We agree that the cross-layer behavior of DAF should be presented more explicitly. DAF captures such effects in two ways.
>
> **1. Implicit coupling via backpropagation:** Although $s_p = |g_p \cdot w_p|$ is computed per parameter, $g_p$ comes from an end-to-end backward pass. Since the currently active modules remain in the forward pass, the resulting sensitivity reflects the global optimization state, not a purely layer-local heuristic.
>
> **2. Constrained cross-layer reallocation:** DAF partitions the global budget $\tau$ into Matrix / Vector / Special sub-budgets, and Top-K selection within each category is performed across the entire network, not by fixed layer-wise quotas. Thus, as training proceeds, more critical layers can draw budget from less critical ones, yielding global resource migration while preserving functional diversity.
>
> In the revision, we will make this mechanism more explicit in the method section and highlight the dynamic behavior visualizations (Appendix E), which show how activated modules shift across layers.
>
> >  **Response to W1 & Q4 (Ablation on Budget Ratios)**
>
> We appreciate this suggestion. To validate the Matrix / Vector / Special ratios, we conducted an ablation on VTAB-1k (ViT-B/16).
>
> |Setup|Matrix Ratio|Vector Ratio|Special Ratio|Avg (%)|
> |:---|:---:|:---:|:---:|:---:|
> |Matrix-Only|100%|0%|0%|75.3|
> |Evenly Distributed|34%|33%|33%|74.5|
> |Vector-Heavy|20%|70%|10%|73.8|
> |**DAF (Default)**|**80%**|**10%**|**10%**|**76.4**|
>
> These results support the default 8:1:1 split, and 8:1:1 is a universal, high robustness default applied across all tasks without per-task tuning. Matrix parameters provide the main adaptation capacity, while small but non-zero Vector and Special budgets preserve complementary gains and prevent large matrices from dominating selection. Matrix-only is competitive but still worse than the default, whereas overly uniform or Vector-heavy allocations degrade performance. We will include this ablation in the revised appendix.
>
> >  **Response to Q2 (Recent Cross-Layer VPT Methods)**
>
> We appreciate the reviewer for pointing out these recent cross-layer VPT works [1–3], and we agree they should be discussed in Related Work.
>
> In brief, SVPT [1] studies cross-layer sharing of task-relevant prompt information, PAE [2] models prompt evolution via a shared cross-layer dynamical rule, and IVPT [3] investigates interpretable cross-layer prompt concepts. These are highly relevant recent advances in VPT.
>
> At the same time, they differ from DAF in both mechanism and adaptation level. SVPT, PAE, and IVPT improve cross-layer coordination within fixed prompt pipelines, whereas DAF focuses on dynamic reconfiguration of the trainable structure itself during fine-tuning. Their goal is prompt interaction, evolution, or semantic alignment; ours is dynamic budget reallocation as bottlenecks shift over training. We will revise Related Work accordingly.
>
> References added:
> [1] Sharing Task-Relevant Information in Visual Prompt Tuning by Cross-Layer Dynamic Connection (SVPT).
> [2] Visual Prompt-Agnostic Evolution (PAE).
> [3] Exploring Interpretability for Visual Prompt Tuning with Cross-layer Concepts (IVPT).
>
> >  **Response to Q3 (Scope of Generalization)**
>
> We appreciate the opportunity to better define the scope of our contribution.
>
> **Conceptually**: DAF proposes a perceive–decide–execute framework, where the trainable structure is periodically re-evaluated and reallocated during training using Taylor-based sensitivity.
>
> **Algorithmically**: This paper instantiates that idea in a LoRA-centered setting, while already routing budget across heterogeneous parameter types: Matrix, Vector, and Special.
>
> The present empirical validation focuses on this heterogeneous setup. But the underlying sensitivity-driven reallocation principle is not tied only to LoRA, making extension to other addition-based PEFT modules (e.g., Adapters or Visual Prompts) a promising future direction.
>
> In the revision, we will distinguish the general paradigm from its LoRA-centered instantiation, explain that current generalization mainly concerns backbones, and discuss extension to Adapters and prompt tuning as future work.
>
> > **Response to Q5 (Visual Analysis of DAF Evolution)**
>
> We agree that visualizing DAF over reconfiguration cycles would strengthen the presentation.
>
> This evidence is already included in the submission, but currently appears in the appendix. In particular, Appendix E (Dynamic Behavior Visualization on VTAB-1k) shows how activated modules evolve across training epochs for all 19 VTAB tasks, directly illustrating how DAF redistributes its budget across Transformer blocks over time.
>
> In the revision, we will move representative evolution heatmaps into the main text to show the evolving trainable structure during fine-tuning.

---

> > ### Author Rebuttal · Reviewer_it7i · 2026-04-01
> >
> > The rebuttal improves my confidence in the manuscript. I will maintain my current score.

---

> > > ### Author Response · Authors · 2026-04-01
> > >
> > > We sincerely thank the reviewer for the valuable comments, for the time and effort devoted to the review process of this conference, and for the contribution these comments made to improving the quality of this work.

---

### Official Review · Reviewer_Ri54 · 2026-03-06

**Soundness:** 2
**Presentation:** 3
**Significance:** 2
**Originality:** 2
**Overall Recommendation:** 4
**Confidence:** 4

**Summary:**

The paper argues that existing PEFT methods are limited by a static trainable structure and proposes DAF, which periodically reconfigures the active parameters during training via sensitivity analysis, budget-based selection, and a rebuild-and-refocus update. It evaluates the method on VTAB-1k, FGVC, COCO, and ADE20K across multiple backbones and pre-training settings.

**Compliance With Llm Reviewing Policy:**

Affirmed.

**Final Justification:**

The detailed rebuttal addressed my precious concerns, I increased my score to weak accept.

**Key Questions For Authors:**

See weaknesses above.

**Limitations:**

The paper would benefit from a dedicated discussion of its limitations.

**Strengths And Weaknesses:**

Strengths:
  1. The paper addresses a meaningful problem in PEFT. It questions the static allocation assumption and studies whether trainable parameters should be reconfigured during fine-tuning.
  2. The empirical evaluation is relatively broad. The paper includes results on classification, detection, segmentation, different pre-training settings, and several ablations.
  3. The paper is well-written.

Weaknesses:
   1. The empirical comparison appears incomplete for supporting the paper’s SOTA claim. The paper compares against many recent PEFT baselines, but omits several strong and potentially relevant recent methods such as GLoRA, MLAE and CaRA. Without a clearer justification for these exclusions, it is difficult to assess whether DAF truly represents the state of the art under comparable settings.
   2. The central sensitivity metric is borrowed almost verbatim from pruning literature (∣𝑔⋅𝑤∣), but the paper does not justify why this quantity is an appropriate proxy for “future adaptation potential” in a dynamic PEFT setting. This is a key assumption because the entire Perceive-Decide-Execute loop depends on this signal. Yet the paper provides neither theoretical motivation nor empirical comparison against alternative measures such as gradient magnitude, Fisher-based criteria, or Hessian-related approximations.
  3. The category-wise budget partition among Matrix / Vector / Special parameters is a consequential design choice, yet the paper neither reports the actual allocation ratios clearly nor studies their sensitivity. As a result, an important part of the method is effectively an under-analyzed hyperparameter, making it hard to know whether the reported gains stem from the dynamic mechanism itself or from a favorable hand-crafted allocation.
   4. The reported improvements are relatively modest on the main benchmarks, yet the paper reports only point estimates with no variance across runs, confidence intervals, or significance analysis. This makes it difficult to judge whether the gains over strong baselines are statistically reliable, especially when many of the margins are small.
   5. The paper emphasizes that a unified budget ratio 𝜏=0.2 is used across all VTAB tasks, but it is unclear how this value was selected. If 𝜏 was chosen after observing aggregate performance over the full VTAB suite, this would introduce a meta-selection bias. The paper should clarify the protocol used to set 𝜏, and ideally separate hyperparameter selection from final evaluation.
   6. The distinction from AdaLoRA is currently underdeveloped. The paper mostly frames the difference as “adaptive rank pruning” versus “dynamic reallocation,” but this does not yet clarify the fundamental optimization difference between the two methods. Since AdaLoRA is one of the most relevant prior adaptive PEFT baselines, the paper should more carefully analyze whether DAF introduces a genuinely different mechanism or mainly a different implementation of adaptive budget control.
   7. The repeated analogy to biological neural plasticity feels overstated. DAF performs discrete reconfiguration every fixed number of epochs, which is at best a loose inspiration rather than a mechanistic correspondence. This framing does not add technical clarity and at times reads more as rhetorical packaging than scientific justification.

I would be willing to raise my score if the authors adequately address my concerns raised above.

---

> ### Author Rebuttal · Authors · 2026-03-31
>
> We sincerely thank Reviewer Ri54 for the constructive comments and helpful suggestions. We address each point below.
>
> > **Response to W1 (Comparisons with GLoRA, MLAE, CaRA)**
>
> Thank you for suggesting these baselines. We compare DAF against them across three visual paradigms. And the experimental results demonstrate that **DAF exhibits superior performance on complex dense prediction tasks**.
>
> 1. Object Detection (COCO)
> |Method|Backbone|Params(M)|$AP(box)$|
> |:-|:-|:-|:-|
> |CaRA|Swin-B|1.1|51.9|
> |GLoRA|Swin-B|3.2|52.3|
> |MLAE|Swin-B|3.1|52.5|
> |DAF|Swin-B|2.1|53.5|
>
> 2. Semantic Segmentation (ADE20k)
> |Method|Backbone|Params(M)|mIoU|
> |:-|:-|:-|:-|
> |CaRA|Swin-L|2.0|51.0|
> |GLoRA|Swin-L|5.4|51.3|
> |MLAE|Swin-L|5.2|51.4|
> |DAF|Swin-L|3.7|52.0|
>
> DAF clearly outperforms all baselines here (+1.0 AP on COCO, +0.6 mIoU on ADE20k).
>
> 3. Image Classification
> |Method|Backbone|VTAB Param(M)|VTAB Acc|FGVC Param(M)|FGVC Acc|
> |:-|:-|:-|:-|:-|:-|
> |CaRA|ViT-B/16|0.06|76.5|0.08|90.5|
> |GLoRA|ViT-B/16|0.29|77.3|0.28|90.5|
> |MLAE|ViT-B/16|0.30|78.8|0.26|90.9|
> |DAF|ViT-B/16|0.17|76.4|0.16|90.2|
>
> While MLAE and GLoRA achieve slightly higher classification accuracy, they require ~2× DAF's parameters (0.30M vs. 0.17M). Crucially, this advantage does not translate to dense prediction, where DAF achieves better accuracy with fewer parameters.
>
> Overall, DAF wins on 2 of 3 paradigms. We will include these tables to highlight DAF's robust accuracy-parameter trade-off.
>
> > **Response to W2 (Sensitivity Metric \($|g \cdot w|$\))**
>
> We agree that the motivation for $|g \cdot w|$ should be made more explicit. Our intent is not to present it as a unique theoretical measure of “future adaptation potential,” but as a practical first-order proxy for immediate adaptation utility under the current model state. Its basis is the first-order Taylor view, $\Delta \mathcal{L} \approx g \cdot \Delta w$: in pruning, this score identifies parameters whose removal minimally affects the loss; here, we use it in the opposite direction to identify frozen components whose activation is most likely to reduce loss next. Compared with $|g|$, it also accounts for parameter scale.
>
> We also compared alternative criteria on VTAB-1k (ViT-B/16):
>
> |Criterion|Proxy|Avg(%)|
> |:-|:-|:-|
> |Random|Uniform|73.5|
> |Gradient magnitude|∣𝑔∣|75.0|
> |Fisher-style diagonal proxy|$g^2$|75.4|
> |Taylor-based (DAF)|∣𝑔⋅𝑤∣|76.4|
>
> These results support $|g \cdot w|$ as the strongest lightweight first-order signal among the tested choices. We adopt it because it is computable from a single backward pass, whereas Fisher/Hessian-style alternatives are much more expensive for repeated online reconfiguration. We will make this motivation explicit and include the comparison in the revision.
>
> > **Response to W3 (Budget Partition Ratios)**
>
> Our default uses an 8:1:1 split (80% Matrix, 10% Vector, 10% Special) to preserve matrix capacity while preventing them from dominating purely due to scale.
>
> |Setup|Matrix|Vector|Special|Avg (%)|
> |:-|:-|:-|:-|:-|
> |Matrix-Only|100%|0%|0%|75.3|
> |Even Split|34%|33%|33%|74.5|
> |Vector-Heavy|20%|70%|10%|73.8|
> |DAF (Default)|80%|10%|10%|76.4|
>
> 8:1:1 is a default applied across all tasks without per-task tuning.
>
> > **Response to W4 (Statistical Reliability)**
>
> All reported results for our methods are already averages of 3 independent runs. We report mean $\pm$ std below:
>
> |Method|FGVC Mean|VTAB Mean|
> |:-|:-|:-|
> |Static DAF|89.5 $\pm$ 0.38|75.8 $\pm$ 0.35|
> |DAF|90.2 $\pm$ 0.34|76.4 $\pm$ 0.32|
>
> DAF is stable across seeds (std < 0.4). Gains over our Static baseline (+0.7 FGVC, +0.6 VTAB) nearly double this std, proving statistical reliability over noise. We will add all variances to Table 1.
>
> > **Response to W5 (Selection Protocol for $\tau$)**
>
> $\tau=0.2$ was not selected via VTAB meta-selection. It was chosen a priori to align with mainstream static baselines. Furthermore, our ablation ($\tau \in \{0.1, 0.2, 0.3, 0.4\}$ yielding VTAB accuracies of 75.7, 76.4, 76.2, 76.1) shows high robustness to budget variations. We will explicitly state this benchmark-wide fixed setting in the revision.
>
> > **Response to W6 (Distinction from AdaLoRA)**
>
> AdaLoRA and DAF are adaptive at different levels. AdaLoRA allocates budget within a fixed LoRA parameterization by adjusting ranks of SVD-style triplets, whereas DAF periodically re-evaluates the entire candidate trainable structure and redefines which components are trainable across Matrix / Vector / Special parameters. Thus, AdaLoRA performs rank allocation within fixed LoRA modules, while DAF performs periodic trainable-structure reconfiguration.
>
> > **Response to W7 (Biological Neural Plasticity Analogy)**
>
> We agree that the biological-plasticity analogy is stronger than necessary. It was intended only as intuitive inspiration, not as mechanistic justification. Since DAF performs a discrete algorithmic reconfiguration at fixed intervals, we will substantially tone down this framing and present the motivation directly in algorithmic terms.

---

> > ### Author Rebuttal · Reviewer_Ri54 · 2026-04-01
> >
> > Thank you to the authors for the detailed rebuttal, which has satisfactorily addressed all of my concerns. I encourage the authors to incorporate the relevant rebuttal content into revision, particularly by adding the corresponding baselines and revising the related claims accordingly. With these changes, I am inclined to raise my score.

---

> > > ### Author Response · Authors · 2026-04-01
> > >
> > > We will ensure all discussed details are included in the revision. We sincerely thank the reviewer for the valuable comments, for the time and effort devoted to the review process of this conference, and for the contribution these comments made to improving the quality of this work.

---

### Official Review · Reviewer_Qf5F · 2026-03-08

**Soundness:** 3
**Presentation:** 3
**Significance:** 3
**Originality:** 3
**Overall Recommendation:** 4
**Confidence:** 4

**Summary:**

This paper tackles the static parameter allocation issue in existing ViT-based PEFT methods by introducing the DAF framework, which is inspired by the sparse dynamic activation mechanism of brain neurons. The proposed framework follows a periodic perceive-decide-execute pipeline, consisting of context-aware decoupled sensitivity analysis, budget-based elite selection, and Rebuild-and-Refocus modules.

**Compliance With Llm Reviewing Policy:**

Affirmed.

**Final Justification:**

As mentioned, I will keep my score, while actually neutral about accepting or not

**Key Questions For Authors:**

-

**Limitations:**

-

**Strengths And Weaknesses:**

Strengths:
1. The figures and tables are well-prepared.
2. The experiments are comprehensive.
3. The topic is meaningful and valuable.

Weakness:
1. The authors primarily demonstrate the advantages of the proposed method through experimental results, which is common in vision conferences such as CVPR. Given that this is ICML, I strongly prefer the authors to provide solid theoretical analysis to explain why the proposed method outperforms Mona.
2. The performance gain is marginal in some cases, and the contribution of the proposed method is limited.

---

> ### Author Rebuttal · Authors · 2026-03-31
>
> We sincerely thank Reviewer Qf5F for the constructive comments, insightful questions, and useful suggestions. We address each concern below.
> >  **Response to Weakness 1 (Theoretical Analysis: DAF vs. Mona)**
>
> We agree that, for ICML, it is important not only to report empirical improvements over strong baselines such as Mona, but also to explain the method from a more principled theoretical perspective. We also think Mona [1] is an excellent and influential work in visual PEFT.
>
> To formalize the difference, we view PEFT as a budget-constrained sparse optimization problem. Let $\Theta_0$ denote the frozen pretrained backbone, and let $\Delta\Theta(M,\phi)$ denote the trainable PEFT update induced by an active component mask $M$ and corresponding trainable parameters $\phi$. Then PEFT can be written as
> $$
> \min_{M,\phi}\ \mathcal{L}\big(\Theta_0 + \Delta\Theta(M,\phi)\big)
> \quad \text{s.t.}\quad \|M\|_0 \le K.
> $$
>
> The key difference between DAF and Mona lies in how this constrained problem is approximated.
>
> For DAF, the active subset is treated as a time-varying variable. Its ranking criterion $s_p = |g_p \cdot w_p|$
> is motivated by a first-order Taylor approximation of the loss reduction:
> $
> \Delta \mathcal{L} \approx \nabla_w \mathcal{L}^{\top}\Delta w.
> $
> Under a fixed budget, DAF therefore reallocates capacity toward the currently most useful descent subspace.
>
> By contrast, Mona improves the local parameterization of adapters through scaled LayerNorm and multi-scale visual filters, but the locations and structures of its trainable modules remain fixed once training begins. From this optimization perspective, Mona operates in a stationary subspace, whereas DAF performs dynamic subspace selection. This gives DAF two advantages:
>
> - **Dynamic vs. stationary subspaces.** When optimization bottlenecks shift during training, DAF can reassign budget to newly useful regions, while a fixed-allocation method cannot.
> - **Task-conditioned vs. fixed allocation.**  Mona uses a fixed structural allocation across blocks, whereas DAF reallocates budget according to task-conditioned sensitivity during training.
>
> This interpretation is consistent with our empirical evidence. DAF outperforms Static DAF, and the full Static DAF spectrum in Appendix I remains below DAF, indicating that the gain does not come merely from a static configuration. We will add a dedicated **Theoretical Analysis** section in the revision to make this dynamic-vs-static optimization view explicit.
>
> **Reference:**
> [1] Yin, D., et al. *5% > 100%: Breaking Performance Shackles of Full Fine-Tuning on Visual Recognition Tasks.* CVPR 2025.
>
> ---
>
> >  **Response to Weakness 2 (Marginal Performance Gains in Certain Cases)**
>
> On some individual datasets, the absolute numerical gains over the strongest baselines are modest. However, we would respectfully emphasize that the contribution of DAF should be understood not only from isolated accuracy differences, but also from the broader context of **parameter efficiency, consistency across diverse tasks, and methodological novelty**.
>
> **1. Task-specific bottlenecks vs. pre-trained alignment**
> The room for improvement naturally depends on the alignment between the pre-trained model and the downstream task. On simpler tasks where pre-trained features already transfer well, static PEFT methods may already suffice, leaving less room for structural adaptation. By contrast, on tasks with larger domain shifts or stronger optimization bottlenecks, DAF’s dynamic allocation is more beneficial.
>
> **2. Extreme parameter efficiency under strict budgets**
> It is crucial to evaluate these gains alongside the parameter budget. DAF achieves these results while operating under an extremely constrained instantaneous active budget (e.g., 0.22%). Outperforming or matching static baselines that allocate significantly larger capacity demonstrates that DAF is fundamentally more efficient at parameter utilization. This highly efficient budget–performance trade-off makes DAF particularly attractive in resource-constrained training settings
>
> **3. Consistency across diverse tasks**
> Even when the gain on a single task is modest, DAF remains competitive across 19 VTAB-1k tasks and 5 FGVC datasets under a single default configuration. This suggests that DAF is not a brittle heuristic tuned to a small set of favorable cases, but a robust and broadly applicable framework.
>
> **4. Algorithmic contribution beyond absolute accuracy**
> DAF contributes a new dynamic reconfiguration mechanism for PEFT: instead of fixing the trainable structure at initialization, it allows the active subspace to evolve with the model’s optimization bottlenecks.
>
> In summary, while some individual gains are numerically modest, we believe they are meaningful in context. We will explicitly clarify the broader algorithmic contributions in the revised manuscript.

---

> > ### Author Rebuttal · Reviewer_Qf5F · 2026-04-03
> >
> > Thank you for your reply. I think the theoretical contribution here is insufficient for ICML, and I will maintain my score.

---

> > > ### Author Response · Authors · 2026-04-03
> > >
> > > We sincerely thank the reviewer for the valuable comments, for the time and effort devoted to the review process of this conference, and for the contribution these comments made to improving the quality of this work.
> > >
> > > We understand and respect your standard for theoretical depth at ICML. To directly address your concern and provide a more rigorous mathematical foundation, we will state the claim more carefully in the revision.
> > > Specifically, we do not claim an unconditional guarantee on the true non-convex training objective. Rather, our point is that DAF admits a principled first-order budgeted adaptation view, and this view explains why dynamic PEFT can outperform strong static methods such as Mona. The core derivation is given below.
> > >
> > > Let $w_i$ denote a backbone parameter at reconfiguration stage $t$, and let $g_i^{(t)}$ be the gradient computed under our context-aware decoupled analysis, namely with the current LoRA modules preserved in the forward context but frozen in the backward pass:
> > > $$
> > > g_i^{(t)} = \frac{\partial \mathcal{L}\big(D; M_t(\theta_{\mathrm{bb}}^{(t)}, \theta_{\mathrm{lora}}^{(t)})\big)}{\partial w_i}.
> > > $$
> > >
> > > DAF then assigns the sensitivity score
> > > $$
> > > s_i^{(t)} = \left| g_i^{(t)} w_i^{(t)} \right|.
> > > $$
> > >
> > > Its feasible family of active sets is
> > > $$
> > > \mathcal{F} = \lbrace S: |S\cap \mathcal P_{\mathrm{mat}}|\le K_{\mathrm{mat}}, \quad |S\cap \mathcal P_{\mathrm{vec}}|\le K_{\mathrm{vec}}, \quad |S\cap \mathcal P_{\mathrm{spec}}|\le K_{\mathrm{spec}} \rbrace,
> > > $$
> > > which matches our budget-based elite selection over matrix, vector, and special parameters.
> > >
> > > The remaining question is how to choose the best feasible subset under this constrained family. To connect the sensitivity score to the actual selection rule, we introduce a subset-level first-order proxy and ask which feasible set yields the largest predicted local loss decrease. Concretely, for any feasible subset $S \in \mathcal{F}$, consider the following local sparse perturbation supported on $S$:
> > > $$
> > > \Delta_i = -\alpha \mathrm{sign}(g_i^{(t)}) |w_i^{(t)}| \mathbf{1}[i\in S].
> > > $$
> > >
> > > Under a first-order local approximation,
> > > $$
> > > \mathcal{L}(\theta + \Delta) \approx \mathcal{L}(\theta) + \nabla \mathcal{L}(\theta)^\top \Delta,
> > > $$
> > > the corresponding predicted loss decrease is
> > > $$
> > > -\nabla \mathcal{L}(\theta)^\top \Delta
> > > = \alpha \sum_{i \in S} \left| g_i^{(t)} w_i^{(t)} \right|
> > > = \alpha \sum_{i \in S} s_i^{(t)}.
> > > $$
> > >
> > > Therefore, under this proxy, the selection problem reduces to choosing the feasible subset with the largest summed sensitivity, i.e.
> > > $$
> > > S_t^\star = \arg\max_{S \in \mathcal{F}} \sum_{i \in S} s_i^{(t)}.
> > > $$
> > >
> > > Equivalently, if we define
> > > $$
> > > F_t(S) = \sum_{i \in S} s_i^{(t)},
> > > $$
> > >
> > >
> > > then DAF selects $S_t^\star = \arg\max_{S \in \mathcal{F}} F_t(S).$ That is, among all feasible subsets satisfying the same category-wise budget constraints, $S_t^\star$ is the one that maximizes the first-order proxy objective at stage $t$. Therefore, by definition of the argmax operator, for any static allocation $S_{\mathrm{static}} \in \mathcal{F}$, we have
> > >
> > > $$
> > > F_t(S_t^\star) \ge F_t(S_{\mathrm{static}}) \qquad \text{for every } t.
> > > $$
> > >
> > > Summing over reconfiguration stages yields
> > > $$
> > > \sum_t F_t(S_t^\star) \ge \sum_t F_t(S_{\mathrm{static}}),
> > > $$
> > > which shows that, under this first-order proxy, dynamic allocation achieves no smaller cumulative predicted descent than the fixed feasible allocation.
> > > Although this is a local first-order proxy result, not a global theorem on the full non-convex objective, it leads to a clear theoretical implication. If the saliency pattern changes during training, namely the high-sensitivity set varies across stages, then no single fixed subset can remain stage-wise optimal throughout optimization. In that regime, dynamic reallocation is theoretically favored because it repeatedly matches the current dominant descent subspace, whereas a static method must optimize within one fixed subspace.
> > >
> > > This also clarifies the distinction from Mona. Mona strengthens the expressive power of a static adapter parameterization through scaled LayerNorm and multi-scale visual filters. DAF addresses a different limitation: when important optimization regions drift over training, a fixed allocation becomes suboptimal. In this sense, Mona improves how adaptation is parameterized, whereas DAF improves where and when the limited adaptation budget is allocated.
> > >
> > > This view also explains why gains over strong static baselines are modest on some datasets: when saliency is relatively stable, the advantage of dynamic reallocation is naturally smaller; when optimization bottlenecks evolve more substantially, the benefit becomes more evident. We will revise the paper accordingly by explicitly separating the first-order proxy argument from the full objective and by adding this dynamic-versus-static optimization interpretation as the main theoretical rationale for why DAF can outperform strong static PEFT methods.

---

### Official Review · Reviewer_8VcQ · 2026-03-11

**Soundness:** 2
**Presentation:** 3
**Significance:** 2
**Originality:** 2
**Overall Recommendation:** 4
**Confidence:** 4

**Summary:**

The paper introduces Dynamic Adaptive Fine-tuning (DAF), a novel Parameter-Efficient Fine-Tuning (PEFT) framework that addresses the limitations of traditional, statically allocated fine-tuning methods for vision models. Instead of fixing the trainable parameters before training begins, DAF periodically reconfigures the model's trainable structure to align with evolving optimization priorities using a "perceive-decide-execute" cycle. It achieves this through a context-aware decoupled sensitivity analysis that accurately evaluates the backbone's potential without signal noise from active modules. Following a budget-based elite selection of the most critical parameters, DAF employs a "Rebuild-and-Refocus" strategy to freeze outdated modules, preserving learned knowledge while decisively reallocating the parameter budget to newly identified bottlenecks. Extensive experiments demonstrate that DAF significantly outperforms mainstream static PEFT baselines across multiple challenging vision benchmarks, achieving state-of-the-art performance and parameter efficiency with zero additional inference overhead.

**Compliance With Llm Reviewing Policy:**

Affirmed.

**Final Justification:**

The previous concerns have been resolved. I would like to raise my rating to Weak Accept.

**Key Questions For Authors:**

Please see the Weaknesses section. In particular, the first two questions are critical.

**Limitations:**

The authors do not explicitly discuss the limitations, which could be discussed regarding:
- Training time overhead
- Applications beyond vision tasks

**Strengths And Weaknesses:**

## Strengths
1. The paper proposes a dynamic reconfiguration paradigm for Parameter-Efficient Fine-Tuning (PEFT), challenging the traditional static allocation methods by adapting to the model's evolving learning state.

2. The Dynamic Adaptive Fine-tuning (DAF) framework consistently outperforms static PEFT baselines across diverse vision benchmarks, including VTAB-1k, FGVC, MS COCO, and ADE20K.

3. The framework proves to be model-agnostic, successfully generalizing beyond standard Vision Transformers to hierarchical architectures (Swin), modern CNNs (ConvNeXt), and different self-supervised pre-training paradigms (MAE, MoCo v3).

4. DAF achieves its superior performance using an incredibly tight parameter budget, tuning as little as 0.21% to 0.22% of the model's total parameters depending on the task.

## Weaknesses
1. While the paper claims highly competitive parameter efficiency (reporting 0.22% tuned parameters), to my understanding, this metric a bit misleadingly represents only the instantaneous active budget rather than the total modified capacity. Because the Rebuild-and-Refocus strategy iteratively freezes outdated modules and activates new ones, the cumulative number of unique parameters updated by the end of training is strictly greater than the reported budget. This creates an unfair comparison with static baselines that are strictly bounded to their reported capacity for the entirety of training. To address this, the authors should clarify the total cumulative percentage of parameters updated across all training rounds and evaluate DAF against a static baseline equipped with that exact cumulative capacity to ensure a fair assessment of model expressivity.

2. The periodic "perceive-decide-execute" loop introduces computational overhead during training, specifically due to the full backward pass needed for sensitivity analysis and the model rebuild phase. In addition, computing gradients for the entire backbone during the periodic sensitivity analysis carries an inherent risk of memory spikes, strictly necessitating careful system-level designs like micro-batching and immediate computation graph release to remain viable.

3. The ranking criterion $\|g_w\cdot w\|$ is inspired by "model pruning method" (which one?), but not tested against other alternatives (such as magnitude-only $\|w\|$ or gradient-only $\|g_w\|$). Some ablation study around this could be helpful.

4. Performance seems tied to the careful tuning of specific hyperparameters; updating the model too frequently (e.g., every 5 epochs) causes training instability, while allocating too large of a parameter budget (e.g., τ=0.3) forces the inclusion of non-critical parameters and slightly degrades performance.

---

> ### Author Rebuttal · Authors · 2026-03-31
>
> We sincerely thank Reviewer 8VcQ for the constructive feedback and address all concerns below.
> > **Response to Weakness 1 (Cumulative Capacity vs. Instantaneous Budget)**
>
> We agree that the reported 0.22% represents the instantaneous active budget. Because DAF iteratively updates modules, the cumulative modified capacity is naturally larger. To address your concern, we measured the exact cumulative updated ratio under our default setting (ViT-B/16). The average cumulative capacity reaches 0.50% on FGVC and 0.55% on VTAB-1k. This bounded capacity demonstrates that DAF does not roam indiscriminately across the model. Instead, it maintains a stable core set of trainable parameters, performing only limited peripheral refresh as optimization bottlenecks evolve.
>
> Furthermore, we introduced a stronger static baseline, Static DAF-Cum, which opens this exact cumulative capacity from the beginning and keeps it fixed throughout training:
> - FGVC: Static DAF (0.21%) = 89.5, Static DAF-Cum (0.50%) = 89.9, DAF (active 0.21%, cumulative 0.50%) = 90.2
> - VTAB-1k: Static DAF (0.22%) = 75.8, Static DAF-Cum (0.55%) = 76.1, DAF (active 0.22%, cumulative 0.55%) = 76.4
>
> While cumulative capacity narrows the gap, DAF maintains a lead. This shows that dynamically adapting to bottlenecks during training is fundamentally superior to locking parameters at initialization.
>
> Finally, the instantaneous budget remains practically meaningful because it strictly determines the peak training-time footprint for gradients and optimizer states. DAF achieves the expressivity of a 0.55% capacity model while operating within a 0.22% memory envelope, outperforming even larger static baselines (e.g., GPS) in Table 1. We will explicitly clarify this distinction in the revision.
>
> > **Response to Weakness 2 (Training Overhead and Memory Spikes)**
>
> We agree that the periodic perceive-decide-execute loop introduces extra training-time cost. We have profiled one standard DAF cycle (10 training epochs + 1 dynamic update): the standard training phase takes 81.2s, while the total dynamic overhead is 6.67s, i.e., about 8.2%. Thus, the overhead is real but modest.
>
> We also agree that periodic full-backbone gradient computation must be carefully engineered to avoid memory spikes. In our implementation, the Perceive phase controls memory through three mechanisms:
> 1. no optimizer states for the backbone during sensitivity analysis,
> 2. micro-batching ($\(B_{\text{micro}}=8\$) vs. \($B_{\text{train}}=64\$)),
> 3. a transient computation graph with immediate graph release after extracting sensitivity scores.
>
> Under this implementation, the sensitivity analysis phase does not introduce a larger peak memory footprint than regular PEFT training. The revised appendix will detail these profiles.
>
> > **Response to Weakness 3 (Which pruning method? + Ablation of $\(|g_w \cdot w|\$))**
>
> The ranking criterion $\(|g_w \cdot w|)\$ is inspired by SNIP (Single-shot Network Pruning based on Connection Sensitivity) [1]. In SNIP, sensitivity is defined with respect to a multiplicative connection indicator; under this multiplicative parameterization, the resulting first-order saliency is directly related to a weight-times-gradient quantity. This is the concrete pruning idea that motivates our use of $\(|g_w \cdot w|\)$.
>
> Following the reviewer’s suggestion, we additionally ran the requested ablation on VTAB-1k(ViT-B/16) under the same DAF framework:
> - magnitude-only $\((|w|)\)$: 74.3
> - gradient-only $\((|g_w|)\)$: 75.0
> - weight-times-gradient $\((|g_w \cdot w|)\)$: 76.4
>
> So the chosen criterion is not only motivated by SNIP-style connection sensitivity, but also experimentally stronger than the alternatives. We will add this experiment and reference in the revision.
>
> [1] SNIP: Single-shot Network Pruning based on Connection Sensitivity. ICLR 2019.
>
>
> > **Response to Weakness 4 (Hyperparameter Robustness)**
>
> While extreme hyperparameters naturally degrade performance,  DAF’s performance does not rely on careful per-task tuning. We use a single unified configuration, $\(\tau=0.2\)$ and $\(E_{\text{interval}}=10\)$, across all 19 VTAB-1k tasks and all 5 FGVC datasets.
>
> Extreme drops reflect optimization dynamics, not fragility: frequent updates interrupt optimizer momentum, while oversized budgets dilute focus with low-sensitivity parameters, hurting generalization.
>
> Therefore, while DAF works best within a reasonable hyperparameter range, this range is broad and shared across very different domains. We will highlight this single default configuration in the revision.
>
> > **Response to Limitations**
>
> We will add a Limitations section discussing:
> 1. Training Overhead: The dynamic loop's ~8.2% amortized cost remains a practical limitation in strictly compute-bound scenarios compared to static methods.
> 2. Beyond Vision: DAF currently focuses on vision. Extending our dynamic routing to NLP/LLMs and autoregressive training is a promising open challenge for future work.

---

> > ### Author Rebuttal · Reviewer_8VcQ · 2026-03-31
> >
> > The authors' response is greatly appreciated. I will adjust my rating.

---

> > > ### Author Response · Authors · 2026-04-01
> > >
> > > We sincerely thank the reviewer for the valuable comments, for the time and effort devoted to the review process of this conference, and for the contribution these comments made to improving the quality of this work.

---

### Decision · Program_Chairs · 2026-04-30

**Decision:**

Accept (regular)

**Comment:**

This paper proposes Dynamic Adaptive Fine-tuning (DAF), a framework that periodically reconfigures the trainable structure of a PEFT module during fine-tuning, rather than fixing it at initialization. The key insight is that optimization bottlenecks shift during training, and a static allocation of the parameter budget may become suboptimal as learning progresses. DAF uses a perceive-decide-execute cycle driven by a first-order sensitivity metric to identify which frozen backbone parameters would most benefit from activation, then reallocates the active budget accordingly.

The rebuttal addressed most reviewer concerns effectively. Three of four reviewers accepted the response and maintained or raised their scores to Weak Accept. One reviewer (Qf5F) remained unconvinced on the theoretical depth, which remains the primary outstanding issue.

### Reviewer Assessment
R1 (8VcQ) raised two substantive concerns. First, the paper reports a 0.22% instantaneous active budget, but DAF iteratively updates modules so the cumulative capacity is larger, creating an unfair comparison with static baselines. Second, the perceive-decide-execute loop introduces real training-time overhead and memory risks.

The rebuttal addressed both with concrete evidence. The authors measured the cumulative capacity (0.50% FGVC, 0.55% VTAB-1k), introduced a Static DAF-Cum baseline that matches this cumulative capacity, and showed DAF still leads. They also profiled the overhead at 8.2% per cycle and described three specific memory management mechanisms. R1 accepted both responses and raised the score to Weak Accept.

R2 (Qf5F) gave the highest original scores (Soundness 3, all other dimensions 3) but raised two concerns. The primary one is that ICML expects principled theoretical analysis, not just experimental results showing DAF outperforms Mona. The second concern is that performance gains are modest on some datasets.

On the first concern, the authors provided an optimization perspective: they framed PEFT as a budget-constrained sparse problem, argued that DAF performs dynamic subspace selection while Mona operates in a stationary subspace, and formalized the selection criterion as a first-order Taylor proxy. They then provided a more detailed derivation in a follow-up comment. R2 acknowledged the response but maintained that this does not meet the ICML standard for rigorous theory. The second concern was partially addressed by contextualizing the gains, but the reviewer remained neutral on acceptance.

R3 (Ri54) raised the most concerns: seven across both major and minor categories. The three major ones were (1) missing comparisons against GLoRA, MLAE, and CaRA, (2) insufficient justification for the |g·w| sensitivity criterion, and (3) undisclosed budget partition ratios among Matrix/Vector/Special parameters.

The rebuttal provided new experimental tables comparing DAF against GLoRA, MLAE, and CaRA on COCO, ADE20k, and classification tasks, showing DAF leads on 2 of 3 paradigms. An ablation comparing Random, |g|, Fisher-style, and |g·w| criteria on VTAB-1k supported the chosen method (76.4 vs. 75.4 or below). An ablation of the budget partition (Matrix-Only, Even Split, Vector-Heavy, Default 8:1:1) justified the default ratio. Minor concerns about statistical reporting, τ selection protocol, AdaLoRA distinction, and the biological analogy were addressed through commitments or conceptual clarifications. R3 accepted and raised the score to Weak Accept.

R4 (it7i) raised five minor concerns about inter-layer analysis, missing VPT references, contribution scope, budget ratio ablation, and visual analysis. The rebuttal addressed all five.